# PRACTICAL HYPERPARAMETER OPTIMIZATION FOR DEEP LEARNING

**Stefan Falkner, Aaron Klein & Frank Hutter**
Department of Computer Science, University of Freiburg, Germany
{sfalkner,kleinaa,fh}@cs.uni-freiburg.de

## ABSTRACT

Recently, the bandit-based strategy Hyperband (HB) was shown to yield good hyperparameter settings of deep neural networks faster than vanilla Bayesian optimization (BO). However, for larger budgets, HB is limited by its random search component, and BO works better. We propose to combine the benefits of both approaches to obtain a new practical state-of-the-art hyperparameter optimization method, which we show to consistently outperform both HB and BO on a range of problem types, including feed-forward neural networks, Bayesian neural networks, and deep reinforcement learning. Our method is robust and versatile, while at the same time being conceptually simple and easy to implement.

## 1 INTRODUCTION

Modern deep learning models depend strongly on the correct setting of many internal hyperparameters. and a practical method for optimizing these models must take into account many desiderata:

**1. Strong Anytime Performance.** Since large neural networks can require weeks to train, hyperparameter optimization (HPO) methods that view performance as a black box function are infeasible. Practical HPO methods should already yield good configurations within a modest budget.

**2. Strong Final Performance.** What matters most at deployment time is the performance of the best configuration a HPO method can find given a larger budget. Since finding the best configurations in a large space requires guidance, this is where methods based on random search struggle.

**3. Effective Use of Parallel Resources.** With the rise of parallel computing, large parallel resources are often available, and practical HPO methods need to be able to use these effectively.

**4. Scalability.** Modern applications of deep learning require the setting of many hyperparameters concerning architectural choices, optimization, and regularization. Practical HPO methods therefore must be able to handle problems ranging from just a few to many dozens of parameters.

**5. Robustness & Flexibility.** The challenges for hyperparameter optimization vary substantially across subfields of deep learning; e.g., deep reinforcement learning systems are known to be very noisy (Henderson et al., 2017), while probabilistic deep learning is often very sensitive to a few key hyperparameters. A practical HPO method should work out of the box in all of these contexts.

There has been much recent progress in the field of hyperparameter optimization. All existing methods have some strengths and weaknesses, yet none of them fulfills all of the desiderata above. Our contribution is to combine the strengths of several methods (in particular, Hyperband (Li et al., 2017) and a robust & effective variant (Bergstra et al., 2011) of Bayesian optimization (Brochu et al., 2010; Shahriari et al., 2016)) to propose a practical HPO method that fulfills all of these desiderata.

## 2 BAYESIAN OPTIMIZATION AND HYPERBAND

**Bayesian optimization (BO)** (Shahriari et al., 2016) has been used to improve the state of the art for several tasks Snoek et al. (2012); Bergstra et al. (2014); Feurer et al. (2015); Melis et al. (2017). To obtain good anytime performance, a recent trend in BO is to extend the traditional blackbox setting by exploiting cheaper fidelities of the objective function (Swersky et al., 2014; Klein et al., 2017a; Swersky et al., 2013; Kandasamy et al., 2017). However, BO is usually based

on Gaussian processes (GPs), which do not typically scale well to high dimensions and – without approximations – exhibit cubic compleixty in the number of data points (scalability); they also do not apply to complex configuration spaces without special kernels (flexibility) and require carefully-set hyperpriors (robustness). Alternative models, such as random forests (Hutter et al., 2011) or Bayesian neural networks (Springenberg et al., 2016; Snoek et al., 2015) scale better with the number of dimensions, but it remains future work to adapt them to the multi-fidelity setting.

**Hyperband (HB)** (Li et al., 2017) is a bandit strategy that dynamically allocates resources to a set of random configurations and uses successive halving (Jamieson & Talwalkar, 2016) to stop poorly performing ones. Compared to Bayesian optimization methods that do not use multiple fidelities, Hyperband showed strong anytime performance, as well as flexibility and scalability to higher-dimensional spaces. However, it only samples configurations randomly and does not learn from previously sampled configurations. This can lead to a worse final performance than model-based approach, as we show empirically in Section 4.

## 3    MODEL-BASED HYPERBAND

We now introduce our new practical HPO method BOHB, named after its components of Bayesian optimization (BO) and Hyperband (HB). We based BOHB's BO component on the Tree Parzen Estimator (TPE) (Bergstra et al., 2011), a robust, scalable, and parallelizable BO method. TPE uses kernel density estimators (KDEs) to model the densities $l(\boldsymbol{x}) = p(y < \alpha | \boldsymbol{x}, D)$ and $g(\boldsymbol{x}) = p(y > \alpha | \boldsymbol{x}, D)$ over the input configuration space instead of modeling the objective function $f$ directly by $p(f|D)$. To select a new candidate $\boldsymbol{x}_{new}$ to evaluate, it maximizes the ratio $l(\boldsymbol{x})/g(\boldsymbol{x})$. Due to the nature of KDEs, TPE is very efficient and supports mixed continuous and discrete spaces.

BOHB relies on HB to determine how many configurations to evaluate with which budget $b$, but it replaces the random selection of configurations at the beginning of each HB iteration by a model-based search. Once the desired number of configurations for the iteration is reached, the standard successive halving (SH) procedure is carried out.

BOHB's selection of a new configuration to evaluate is detailed in Algorithm 1. In order to keep the theoretical guarantees of HB, this also includes sampling a constant fraction $\rho$ of the configurations uni-

---

**Algorithm 1:** BOHB's sampling procedure

**input** : observations $D$, fraction of random runs $\rho$, percentile $q$, number of samples $N_s$, min number of points in a model $N_{min}$

**output:** next configuration to evaluate

with probability $\rho$ **return** random configuration (1)

$b = \max_{b'} \{b' : |D'_b| \geq N_{min} + 1\}$  (2)

**if** $b = \emptyset$ **then  return** random configuration  (3)

$\alpha = q^{\text{th}}$ percentile of all observations in $D_b$  (4)

fit KDEs for $l(\boldsymbol{x})$ and $g(\boldsymbol{x})$ on $D_b$ (see text)  (5)

draw $N_s$ samples $\sim l(\boldsymbol{x})$  (6)

**return** sample with highest ratio $l(\boldsymbol{x})/g(\boldsymbol{x})$  (7)

---

formly at random (line 1). Otherwise, the largest budget with at least $N_{min} + 1$ observation (line 2) forms the basis for building a model (lines 4-5) from which new candidates are sampled (lines 6-7). If no budgets contain enough data points, a random configuration is sampled (line 3).

We parallelize BOHB as follows. We start with the first iteration that sequential HB would perform (the most aggressive one, starting from the lowest budget). We sample configurations with the strategy outlined in Algorithm 1, and only start the next iteration in parallel once the currently-running ones do not require enough concurrent evaluation to keep the available workers busy. We note that HB has also been parallelized in independent work (Li et al., 2018) where each iteration is executed by a separate pool of workers, rather than joining all workers into a single pool and preferentially execute runs with smaller budgets. Our strategy (a) yields better speedups by using all workers in the most aggressive (and often most effective) iteration first, and (b) takes full advantage of models built on smaller budgets. Figure 1 (top left) demonstrates that this parallelization approach can effectively exploit many parallel workers.

## 4    RESULTS AND CONCLUSION

We evaluated the empirical performance of BOHB on different tasks: optimization of feed forward neural networks, Bayesian neural networks (BNNs), and deep reinforcement learning agents. We

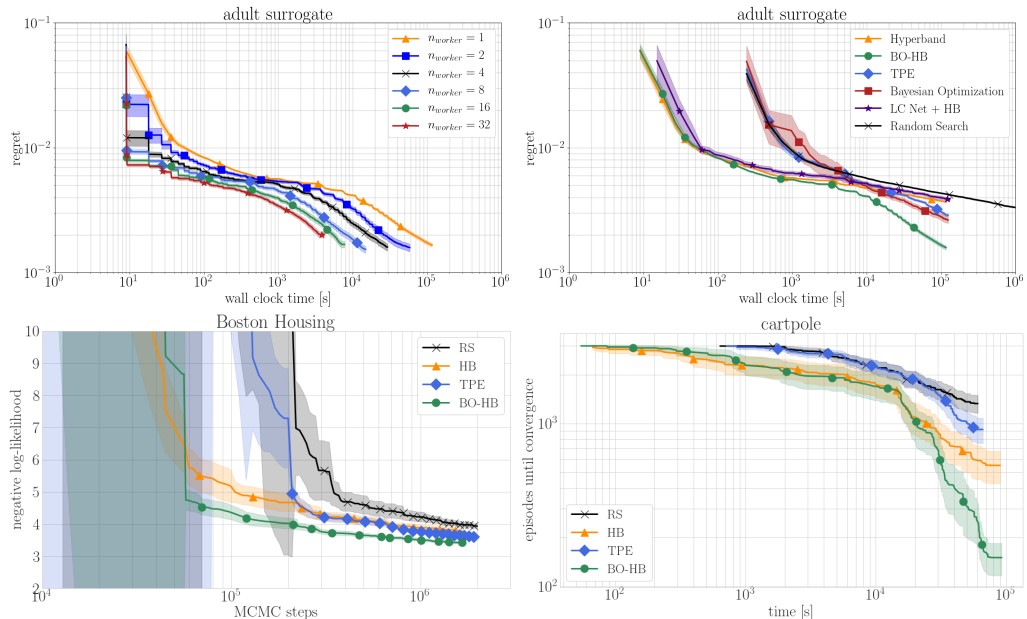

Figure 1: The top left panel shows results for BOHB on a surrogate benchmarks with different numbers of workers, demonstrating the effective parallelization. The other plots compare BOHB to random search (RS), TPE, and HB on the same surrogate benchmark (top right), the *BNN on Boston housing* task (bottom left) and the reinforcement task *cartpole*. The results show that BOHB starts out strong, like HB, but converges more quickly to the optimum, like TPE. The plots show the mean and 2 standard error of the mean based on 512 runs (top row) and 50 runs (bottom row).

summarize the results in Fig. 1. We used the TPE implementation from Hyperopt (Bergstra et al., 2011), and the GP-based BO implementation as well as the BNN from RoBO (Klein et al., 2017b).

For the feedforward network, we constructed a surrogate benchmark following Eggensperger et al. (2015); optimizing a surrogate instead of the real objective function is substantially cheaper, allowing us to afford many independent runs and draw statistically meaningful conclusions. We optimized six hyperparameters and architectural choices: initial learning rate, decay factor, batch size, dropout, number of layers, and units per layer. As budgets for BOHB, we used different training times. We show results for the Adult (Kohavi, 1996) dataset; they were qualitatively similar for 5 other datasets.

For the BNNs, we optimized the hyperparameters and the architecture of a two-layer fully connected BNN trained with stochastic gradient Hamiltonian Monte-Carlo sampling (SGHMC) (Chen et al., 2014) with scale adaption (Springenberg et al., 2016). We considered the UCI (Lichman, 2013) regression dataset *Boston housing* and report the negative log-likelihood of the validation data. The hyperparameters are the step length, the length of the burn-in period, the number of units in each layer, and the decay parameter of the momentum variable. The budget here were MCMC steps.

In the third example, we optimized eight hyperparameters of proximal policy optimization (PPO) (Schulman et al., 2017) to learn the *cartpole swing-up* task. For PPO, we used the implementation from the TensorForce framework (Schaarschmidt et al., 2017); for the cartpole environment we used the implementation from OpenAI Gym (Brockman et al., 2016). The objective function to minimize was the mean convergence time of nine individual trials with different seeds. As budgets for HB and BOHB, we used less trials as low-fidelity approximations.

The three benchmarks in Fig. 1, as well as several others we studied but omitted due to space constraints, all show similar behavior: HB and BOHB perform well early on, because of their cheaper low-fidelity observations compared to TPE and RS. With a large enough budget, TPE typically overtakes HB, but BOHB consistently performs better. We note that in the adversarial case, i.e., for very large budgets, or when performance on low budgets is misleading, BOHB's subsampling may waste resources and BOHB can be slower than pure TPE by a constant factor.

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
