# OpenReview forum: "Practical Hyperparameter Optimization for Deep Learning"
_ICLR.cc/2018/Workshop — Accept_

### Official Review · AnonReviewer3 · 2018-02-21
**Useful but perhaps somewhat obvious**

**Rating:** 7
**Confidence:** 5

**Review:**

This paper proposes combining Bayesian optimization (BO) with the Hyperband algorithm of Li et al 2017.  Specifically, they replace the random configurations of Hyperband with ones chosen by the tree Parzen estimator BO method.  I think this idea is rather obvious making the novelty of the work limited, but the resulting algorithm is clearly useful and has not been explicitly suggested or tested in literature before.  As such, it makes a good fit for a three page short that will be helpful to the community, even if there is little it the way of new ideas.

To explain more specifically, when I read the original Hyperband paper (https://arxiv.org/abs/1603.06560v1) I was infuriated by their overclaiming and criticism of BO in the context of missing the seemingly obvious point that forcing BO to always carry out a full evaluation is spurious and suboptimal (see quote from private correspondence at the bottom of the review).  I doubt I was the only one to make this assessment at the time and so I am a little surprised that it has taken this long for something to appear in the literature making this seemingly obvious point that there is no reason that Hyperband should choose configurations randomly rather than in principled, BO-based manner.  However, I have not actually seen anything making this point in the literature until the reviewed paper; it is certainly to the best of my knowledge the first paper to the provide numerical results on using Hyperband with principled configuration choice.  The resulting algorithm unsurprisingly outperforms Hyperband and thus provides clear utility to the community.  Consequently, I think the paper should be accepted, even though I also feel it is quite limited in terms of novelty and technical contribution.

Even though the paper is well-written at a low level, well motivated, and easy to follow, I do feel that the relative prioritization of space could be improved.  Namely, I think more space needs dedicating to explaining the specifics of the proposed approached and that this could be found by condensing the other sections.  In particular, the role of b in algorithm 1 should be explained.

Pros:
- Useful algorithm
- Clear contribution
- Impactful research area
- Good experimental results clearly presented

Cons:
- Key idea very straightforward
- Some key specifics of the proposed approach somewhat glossed over


Specific errata
- q is not actually used in Alg 1
- b is not really explained and D_b not defined in Alg 1
- Should eta in the last line be rho?

Quote from private correspondence on Hyperband paper

"The most damning part of [the Hyperband paper] is that there is nothing in the existing BO schemes to stop you from also changing the budget spent on each point... Instead one should use partial evaluation of points with revisiting in the same way that they do.  There are theoretical results that show ... one should spend the minimum possible budget on each evaluation and maximize the number of evaluations you make... For GP based BO, the cubically increasing cost of the BO itself is so high that setting this budget remains an open and active research question, but for SMAC and TPE which have negligible overheads, you can comfortably just set it low."

---

### Official Review · AnonReviewer2 · 2018-03-12
**Combination of two hyperparameter optimization methods**

**Rating:** 7
**Confidence:** 3

**Review:**

The paper proposes to combine Tree Parzen Estimator (TPE) (Bergstra et al., 2011) and Hyperband (Li et al., 2017) for hyperparameter optimization. The proposed method uses TPE as one step of Hyperband that samples hyperparameter configurations. The method is evaluated on three tasks and seem to outperform both TPE and Hyperband.

More extensive experimental section would be needed for a full paper but this should be good enough as a workshop paper.

Some improvements that could be made:
- Algorithm 1 is unclear. What is b and D_b? Is the argmax over a family of datasets?
- The first use of the acronym SH in the last paragraph of Section 3 doesn't contain its long form.
- Legend of the top row of Figure 1 is difficult to read in the printed version.

---

### Decision · Program_Chairs · 2018-03-20
**ICLR 2018 Workshop Acceptance Decision**

**Decision:**

Accept

**Comment:**

Congratulations, your paper was accepted to the ICLR workshop.